# Characterization of Thermal Gradient Effects on a Quartz Crystal Microbalance

**DOI:** 10.3390/s22197256

**Published:** 2022-09-24

**Authors:** Marianna Magni, Diego Scaccabarozzi, Ernesto Palomba, Emiliano Zampetti, Bortolino Saggin

**Affiliations:** 1Rebel Dynamics, Via Carlo Porta 38, Cesana Brianza, 23861 Lecco, Italy; 2Mechanical Engineering Department, Polytechnic University of Milan, Via La Masa 1, 20156 Milano, Italy; 3National Institute for Astrophysics INAF-IAPS, Via del Fosso del Cavaliere 100, 00133 Roma, Italy; 4Institute of Atmospheric Pollution Research, National Research Council of Italy, 00015 Monterotondo, Italy

**Keywords:** QCM, TGA, CAM, calibration, uniform temperature, thermal gradient, frequency variation

## Abstract

Quartz crystal microbalances are widely used sensors with applications for the detection of very-low-mass deposition in many different fields, from contamination monitoring in the high vacuum of deep space missions to the monitoring of biological activity or pollution using specifically designed active substrates. These sensors are very stable over time; nevertheless, their sensitivity to the temperature is well known, and different implementations have been devised to correct it, e.g., through compensation with a dual crystal. This paper deals with the effects of temperature on QCM but separates the case of uniform crystal temperature from the case of in-plane temperature gradients considering a QCM based on quartz crystals with deposited film resistors used as both RTDs and heaters. This configuration allows both an accurate temperature measurement and efficient thermal control, allowing the achievement of crystals temperatures in the order of 400 °C higher than the environment with a low power dissipation of the order of 1 W. The film resistors deposited around the electrodes allow directly measuring the average crystal temperature and directly delivering power to the crystal for thermal control. The localized delivery of the heat nevertheless also determines uncommon temperature fields on the crystal, and thus, an analysis of both the effects of temperature on the new microbalance was performed. The temperature gradient has strong effects on the frequency; therefore, along with the temperature, the thermal gradients have tobe compensated. The calibration of the QCM thermometers and the assessment of the achievable measurement accuracy were performed, as well as the determination of the frequency–temperature relationship. The comparison between frequency changes in the case of uniform temperature and those observed while using crystal heaters proved that temperature gradients have a strong effect on the crystal frequency. To identify the temperature field on the crystal surface of a QCM crystal, the gold coating of the deposited films was removed to achieve an emissivity acceptable for thermal imaging with an IR camera. Moreover, image processing for emissivity correction was developed. In order to correlate the temperature gradient with the frequency variation, a test campaign was performed to measure the frequency changes derived from different power levels delivered to the crystal heaters. From this test campaign and thermal analysis, the effect of the thermal gradient was assessed.

## 1. Introduction

QCMs (quartz crystal microbalances) can provide continuous monitoring of very-small-mass deposition with high stability over time [1]; this kind of sensor, therefore, finds applications in many different fields, i.e., space [2], chemical and pharmaceutical industries [3,4,5] and environmental monitoring. The physical and mechanical properties of quartz crystals are quite stable with respect to environmental conditions. Moreover, the measurement is not influenced by gravitational forces [6], ambient pressures have a small effect on measurements and best performances are achieved under high-vacuum conditions, and the operating temperature range has no lower limit [7]. Moreover, the typical masses of these instruments are in the tens-of-grams range, and there is no trade-off between size (collecting area) and sensitivity, so miniaturization is possible. The above characteristics make QCMs optimal candidates for space usage. The possibility of the active thermal control of the crystal enables performing thermogravimetric analyses (TGA), a common technique for in-ground applications that would be very interesting for planetary exploration [8]. Given these perspectives, in 2014, under an ESA contract, our group developed the Contamination Assessment Microbalance (CAM) instrument, a QCM with potential usage for the contamination assessment of space hardware, both during ground activities and during flight [9,10,11,12].

The CAM instrument is based on the double-crystal configuration, i.e., with a measuring crystal and a reference one, and the mass deposition measurement is based on the beating frequency. This configuration allows reducing the frequency range to be measured because the temperature-generated drifts are compensated. The configuration of the crystals developed for this instrument would, nevertheless, allow for the accurate compensation of the temperature effects when using the single-crystal configuration because the temperature is directly measured on the crystals through a deposited-film thermal resistor. This is quite relevant in space applications, where the thermal environment can jeopardize the temperature compensation of the double-crystal configuration because of varying heat fluxes (sun or planetary) on the exposed crystal. CAM crystals include film resistors deposited on each face, and they can be used either as temperature sensors or as heaters. This film heater is the most effective way to increase the crystal temperature because the power is released right at the controlled point; actually, with a power in the range of 1 W, a difference in the order of 300 °C from the environment can be achieved [10]. Nevertheless, the drawback of the direct delivery of power to the crystal is an increase in the thermal gradients. While the sensitivity of QCMs to temperature is well known and also the object of recent studies [13,14,15], thermal gradients are mostly neglected.

The effect of non-uniform temperature distributions over the crystal surface has been analyzed and modeled in the literature in the last century [16,17,18]. Nevertheless, the prediction of the effect is awkward because it depends on many temperature functions of the crystal’s mechanical and physical properties. The effect of temperature gradients, therefore, requires an experimental characterization similar to that performed to characterize the effect of uniform temperature.

This study was aimed at characterizing the effect of temperature gradients, and it was carried out following the classical approach for linear systems, relying on the superimposition of the effects. The issue of the sensitivity to temperature gradients was analyzed through an experimental campaign allowing the characterization of the quartz crystal temperature distribution along with its natural frequency. The effect of temperature gradients was eventually determined by removing the effect of the average temperature from the combined results of the temperature and the gradients.

In Section 2, the calibration of the frequency in a uniform temperature field under vacuum conditions is reported, whereas Section 3 describes the thermal gradient identification on the surface of the crystal. Section 4 finally concludes the paper.

## 2. Quartz Crystal Temperature Sensitivity

The determination of the relationship between oscillating frequency and temperature was performed between 20 °C and 100 °C in a vacuum chamber with an internal pressure of 0.1 mbar (with a measurement uncertainty of ±15% of the reading), achieved by a rotary vacuum pump (Varian DS 402). In order to ensure the cleaning of the chamber and avoid contamination during the calibration, the chamber was cleaned with isopropyl alcohol before testing. The microbalance was mounted on a cryostat placed over a thermoelectric cooling element (TEC). The latter allows for the heating and cooling of the crystal assembly. To measure the temperature of the TEC during the tests, a platinum temperature sensor (Pt100 type, accuracy class A) was used. A cold-water loop was used as a heatsink for the TEC, and to measure the crystal temperature, the embedded deposited sensor was calibrated. Figure 1 shows the measured electrical resistance of the deposited heater (in red) and the temperature sensor (in blue) between −20 °C and 100 °C.

Least-Square (LS) linear fitting was extracted from the measured data. The obtained results evidenced good linearity of the deposited resistors, providing a sensitivity of 7.400 × 10^−2^ Ω/°C and 7.498 × 10^−2^ Ω/°C for the temperature sensor and heater, respectively. The standard deviation of the regressors was found to be constrained to 0.05% of the measured slope. Recently, the good performance of this type of deposited resistor was proved even at low temperatures, down to liquid nitrogen temperature [19]. The setup to evaluate the temperature sensitivity of the crystal is shown in Figure 2a, where the positioning of the Pt100 sensors, cryostats, and TEC is provided. The testing procedure required three different cycles within the investigated temperature range. Figure 2b depicts the performed cycling. The TEC was used in reverse mode during heating and normal mode when cooling was required.

A flowchart of the crystal calibration procedure is shown in Figure 3.

The measured variation in the crystal frequency during calibration is shown in Figure 4, where three VC cycles are compared.

Comparing the measured trends, it was found that the measurement repeatability (1σ band) was about 12.6 Hz. In fact, despite the curve is similar for the various cycles and not far from the theoretical one [7] for AT-cut crystals, analyzing the measured trends in detail, it can be shown that the measured frequency variation is characterized by a negative drift of the frequency curves with time, evidencing that some condensation was occurring. This problem is related to the chamber cleanliness and the achieved vacuum level, which, being in the range of a few Pa, does not warrant the removal of the outgassed components that at least partially also condense on the crystal. Nevertheless, the obtained result was judged to be accurate enough for the intended application, and it was decided to keep the contamination effect as part of the procedure uncertainty. Thus, a third-order polynomial was extracted from the measured data to model the frequency change vs. temperature in uniform temperature conditions. The best fitting result considering all three cycles is given in Table 1.

The regression standard deviation was found to be 13 Hz, therefore providing an uncertainty of less than 2% of the full-scale effect in the considered temperature range.

## 3. Thermal Gradient Measurement

### 3.1. Infrared Camera Calibration

The temperature distribution on the crystal surface was identified by using a micro-bolometric infrared camera (NEC TH7102). Thermal imaging leads to huge temperature uncertainty when there is poor knowledge of the surface’s emissivity, such as the one achievable through tabulated figures. This is especially relevant in the case of low average values, where relative uncertainty can be quite large. To achieve acceptable uncertainty, the emissivity was firstly measured nevertheless, it was also necessary to calibrate the infrared camera in order to overcome the declared accuracy of 2 °C and finally to determine the different emissivities of the imaged materials. Camera calibration was performed by using a black body (BB) (model type CS 110, DIAS). The black body consists of a box with an aperture that behaves like an ideal emitter, providing a temperature uncertainty of 0.3 °C and nominal emissivity of 0.98. In order to determine the corrective factor, the temperature of the black body was varied between 40 °C and 100 °C with steps of 5 °C, and for each value, a thermal image of the black body was taken. The test was repeated three times to consider the measurement repeatability. The measurement setup is shown in Figure 5.

The experimental procedure consisted of the following phases:Setting the temperature to the black body and waiting until the steady-state condition was reached;Taking the thermal image of the BB by using the infrared camera with a set emissivity of 1;Analyzing the thermal image and evaluating the average, minimum, and maximum temperatures within the area of the BB using postprocessing software (InfraRec analyzer) before computing the corrective factor *Fc*.

Fc is defined as the ratio between the radiated intensity of the calibration BB and that measured by the camera, i.e.:(1)$$Fc=\frac{0.98\;T_{BB}^{4}}{T_{TH}^{4}}$$
where *T_BB_* and *T_TH_* are the set temperature of the black body and the average temperature measured by the infrared camera (in Kelvin), respectively. Figure 6 shows the experimentally determined *Fc* values and the points corresponding to the third-order polynomial regression of those data.

The third-order polynomial model obtained by LS fitting is provided in the following:(2)$$Fc\left(T\right)=4.71\times 10^{-7}\;T^{3}-1.08\;T^{2}+8.31\;T+0.802$$
where *T* is the temperature measured by the infrared camera in °C units. The uncertainty for the *Fc* factor was computed as well from the residual sum of squares, obtaining a value of 0.003. Thus, the measured temperature-dependent corrective factor was applied to the measured temperatures during the emissivity measurements of the materials and the thermal gradient assessment. The variable of utmost interest is the temperature distribution on the electrode whose surface was originally gold-coated, i.e., with emissivities in the range of 2–5%; the gold film plating was removed to allow for the recording of meaningful temperature mappings. For the emissivity determination, the crystal was thermally connected to a heated plate through a thermal filler to ensure matching between the crystal and plate temperatures. The measured emissivities for the electrode and quartz were 0.254 and 0.629, respectively, with a measurement uncertainty of about 0.007 for both materials.

### 3.2. Setup Description

Having characterized the emissivity, the infrared camera was used to determine the thermal fields on the crystal while it was heated by the built-in heater. To determine the thermal gradient, firstly, the crystal was mounted on a structure that allows connecting the deposited resistances to the devices. The structure with the cabling connection is shown in Figure 7.

There are two deposited films on the crystal, one on each side. One film is used as a heater, and the other serves as a temperature sensor. A circuit connected to the two electrodes allows measuring the crystal frequency, whereas the temperature sensor is connected to a Keysight 34,970 A multiplexer that measures the resistance with the four-wire technique. Finally, the bottom heater is fed by a voltage-controlled power supply. The IR camera was placed perpendicularly to the surface of the crystal at the minimum focus distance of 35 cm, and a cardboard cylinder was used to avoid the influence of the laboratory background variable because of moving people and heat sources. The background temperature was measured with three platinum resistance thermometers (PT100, accuracy class A) placed along the cylinder and acquired by the multiplexer.

### 3.3. Thermal Gradient Measurement

The test was performed by providing a voltage of 20 V (0.022 A) to the heater from the power supply. Once the steady state was achieved, the thermal image was taken and analyzed by imposing the computed material emissivity and the thermal imager’s corrective factor. The measured temperature map and detailed views of the electrode and heater areas are shown in Figure 8.

As shown by the thermal map, the temperature ranges between 96 °C and 80 °C in the electrode area. Moreover, a gradient of temperature toward the crystal center is found, as testified by the brighter area in the middle of the image. To compute the temperature uncertainty, propagation according to ISO-GUM was performed:
(3)$$T_{c}=\sqrt[{4}]{\frac{{\left(T_{th}F_{c}\right)}^{4}-\left(1-\varepsilon_{c}\right)T_{bk}^{4}}{\varepsilon_{c}}}$$
(4)$$u_{Tc}=\sqrt{{\left(\frac{\partial T_{C}}{\partial T_{th}}u_{Fc}\right)}^{2}+{\left(\frac{\partial T_{C}}{\partial T_{bk}}u_{Tbk}\right)}^{2}+{\left(\frac{\partial T_{C}}{\partial \varepsilon_{c}}u_{\varepsilon_{c}}\right)}^{2}}$$
where *T_c_* is the corrected temperature, *F_c_* is the IR camera calibration function, *T_th_* is the temperature indicated by the IR camera, *T_bk_* is the background temperature, and *ε_c_* is the electrode emissivity, while *u_Fc_* is the uncertainty of the calibration function, and similarly, the other “*u*” variables indicate the uncertainties of all parameters. The results of temperature correction for the electrode and the heaters are summarized in Table 2, along with the corresponding measurement uncertainties.

### 3.4. Thermal Gradient Effect on The Frequency Variation

In order to identify the temperature gradient effects on the frequency variation, a test was performed by feeding the deposited heater with the power supply at different voltages (from 3 V to 24 V, with steps of 3 V each). The acquisition was completed following the steps described below:The heater was powered at constant voltage for a time sufficient to reach the steady-state condition;The electrical resistance of the temperature sensor film and the temperatures of the background were simultaneously measured with the multiplexer;The frequency was recorded by using a frequency counter synchronized with the multiplexer unit.

Figure 9 provides the block diagram of the measurement chain.

The thermal gradient characterization was performed in both heating and cooling conditions by increasing and decreasing the feeding voltage, respectively. For the calibration of the crystal frequency vs. uniform temperature, the setup in Figure 7 was mounted in the vacuum chamber.

The analysis was carried out by determining the mean temperatures of the electrode and heater areas (as conducted for the thermal gradient measurement) from the acquired thermal images and measuring the frequency variation as the difference between the initial value (i.e., at ambient temperature) and the values related to the different heating conditions. Finally, the temperature of the electrode was measured by determining the electrical resistance of the calibrated temperature sensor deposited on the crystal. Figure 10 shows the obtained frequency variation vs. the electrode temperature. In the following equation, the LS fitting of the experimental data by using a third-order polynomial function is given:(5)$$Frequency=3.5\times 10^{-3}\;t^{3}-0.5397\;t^{2}+52.73\;t-1044\;\left(\mathrm{Hz}\right)\;$$
where *t* is the measured crystal temperature in °C. The computed RMSE from the fitting was 87.1 Hz.

It can be observed that all heating and cooling cycles at different power levels are quite overlapping, evidencing the repeatability of the effect of the thermal gradient on the oscillating frequency. Moreover, during the test, the crystal was always warmer than the environment, so the condensation on it was negligible, despite the vacuum chamber used in the tests being the same used in the uniform temperature characterization. The uniform temperature–frequency variation trend was compared with the obtained results. It can be seen that already at low temperatures, i.e., around 40 °C and 50 °C, the measured difference between the uniform and non-uniform conditions matches the maximum variation measured during the crystal calibration at a uniform temperature. The difference becomes larger with increasing temperature, achieving about a 1600 Hz difference at 100 °C.

The huge difference in the sensor output between the uniform temperature and the localized heating conditions recorded in these tests demonstrates the prevailing effect of the thermal gradient and poses new objectives for QCM development related to the correction of the effect of the thermal gradient. For that purpose, a finite element model of the crystal is under development to numerically derive the thermal gradient on the electrode and find a correction procedure to increase the measurement accuracy provided by the microbalance when operating in non-uniform temperature conditions.

It must be noted that the double-crystal configuration would also compensate for the effect of temperature gradients if the temperature distribution was the same on both the measuring and reference crystals. The remaining error once again depends on the heat fluxes on the exposed crystal, which, in space applications, is often a variable parameter in different mission phases.

## 4. Conclusions

The quartz crystal under development is characterized by an innovative crystal configuration that includes heaters and thermometers deposited on the surfaces as film resistors. The great advantages of the new configuration in terms of the accuracy of the temperature measurement and the efficiency of thermal control raise new issues related to significant thermal gradients on the crystal. The temperature–frequency characterization of the quartz crystal microbalance required not only the analysis of the effect of different uniform temperatures of the crystal but also the identification of the effect of the temperature gradients. The built-in temperature sensor allowed measuring the crystal temperature right at the sensing area, i.e., on the electrode border, with minimum error. The direct measurement of the crystal temperature allows for performing an accurate determination of the relationship between the oscillating frequency and temperature. Crystal characterization was performed within a temperature-controlled enclosure in a vacuum chamber, and the obtained curve qualitatively matches the literature data. Compatibility with the literature is not expected because of the dependency of this characteristic on the specific crystal manufacturing details. The test also showed the high sensitivity of the instrument to the contamination that was noticed in the medium vacuum of the test chamber.

The analysis was then focused on the crystal frequency behavior under temperature gradients. The results of the combined temperature and thermal gradient effects on the frequency variation were compared with the uniform temperature–frequency variation relationship. It was evidenced that, with the adopted configuration, the effect of thermal gradients when directly heating the crystal with the built-in heater is by far more relevant than that of the average temperature. The temperature field on the crystal surface was measured with a thermal mapper, which allowed determining the temperature distribution and the thermal gradients at the electrode border. The information, nevertheless, was affected by the low spatial resolution of the thermal image. In the next steps of the research, a thermal model of the crystal surface will be developed to predict, with high spatial resolution, the temperature field on the quartz crystal surface and, as a result, the thermal gradients on it. It is mandatory to implement a procedure for the thermal gradient correction, a key step toward achieving a deeper knowledge of the sensor behavior and, eventually, more accurate mass measurements.

## Figures and Tables

**Figure 1 sensors-22-07256-f001:**
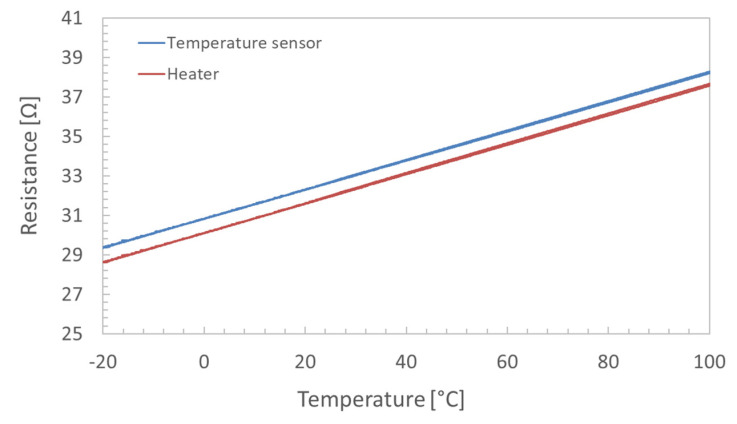
Calibration of the deposited resistors on the QCM in a thermal bath.

**Figure 2 sensors-22-07256-f002:**
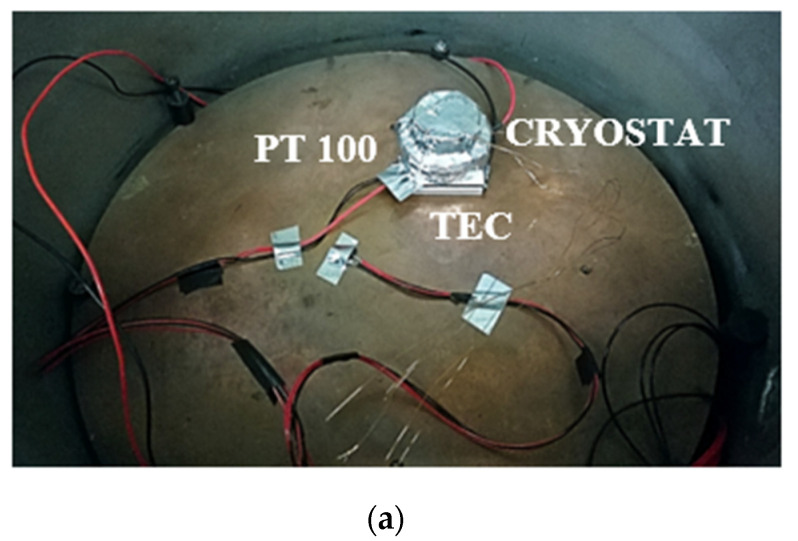
(**a**) Measurement setup for microbalance calibration in vacuum; (**b**) measured temperatures of the crystal, heat sink, and TEC.

**Figure 3 sensors-22-07256-f003:**
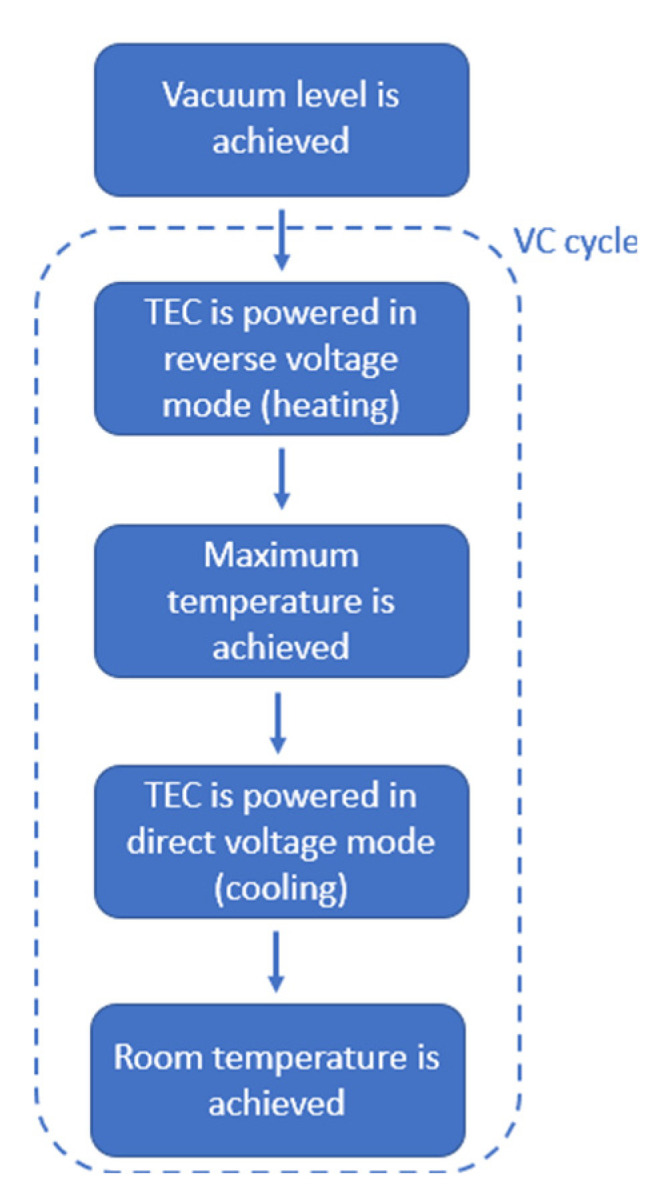
Flowchart of the calibration procedure.

**Figure 4 sensors-22-07256-f004:**
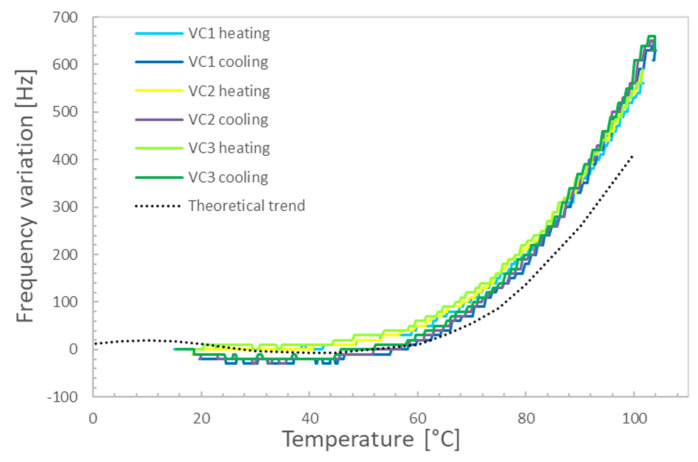
QCM calibration in a uniform temperature environment: results of three subsequent cycles.

**Figure 5 sensors-22-07256-f005:**
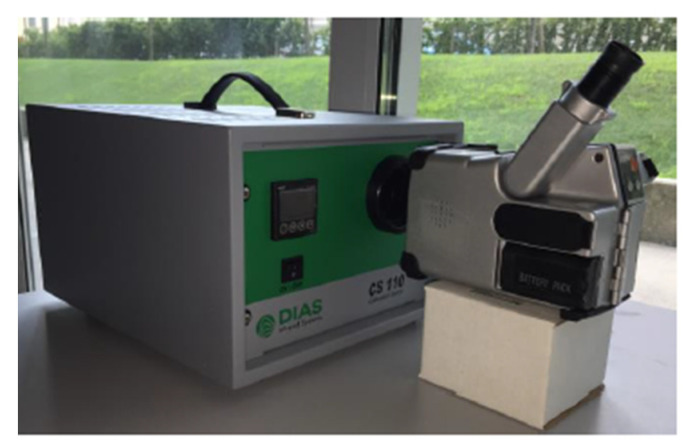
Calibration setup with relative positioning between the camera and the calibration body.

**Figure 6 sensors-22-07256-f006:**
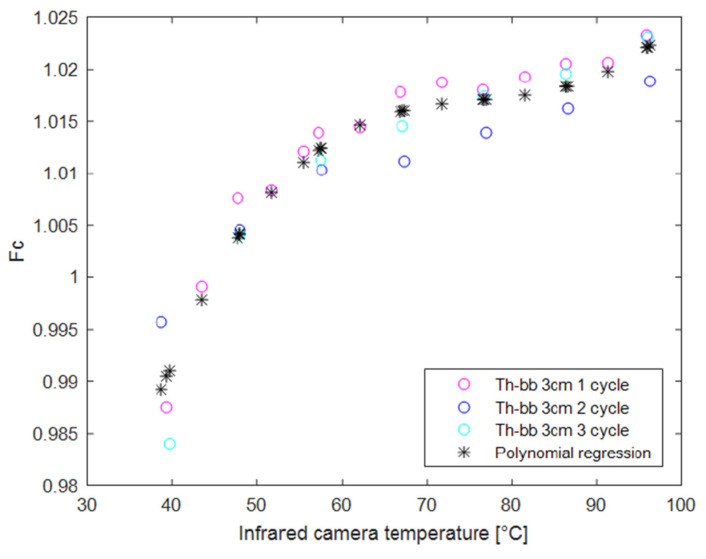
Measured corrective factor *Fc* during infrared camera calibration vs. *T_TH_* in °C. Three different tests (named Th-bb) at a distance of 3 cm are provided.

**Figure 7 sensors-22-07256-f007:**
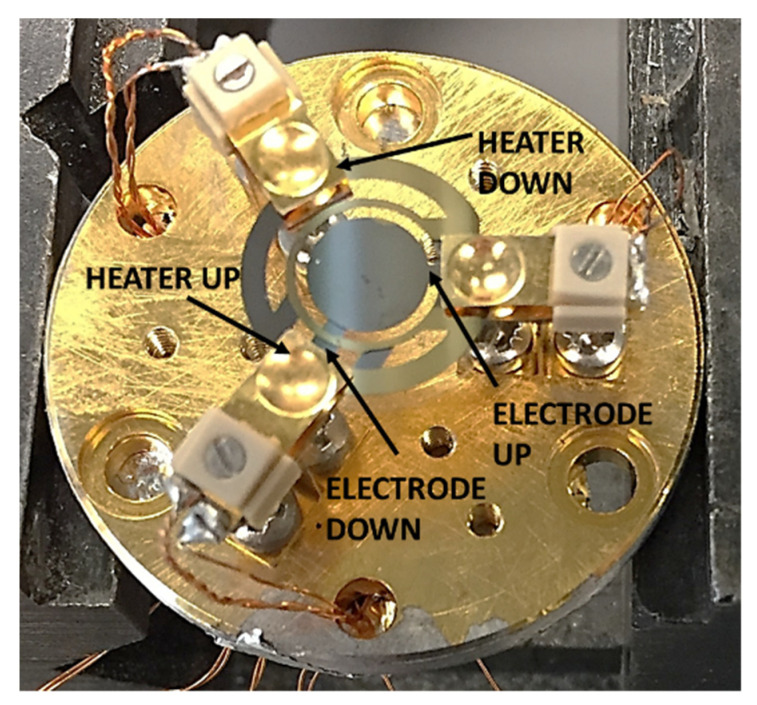
Measurement setup for microbalance calibration.

**Figure 8 sensors-22-07256-f008:**
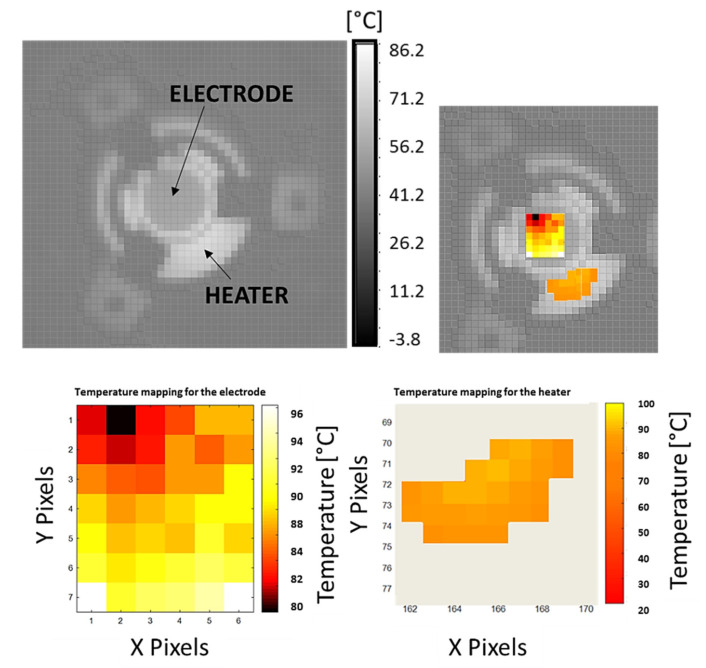
Temperature map and detailed views of the electrode and heater areas.

**Figure 9 sensors-22-07256-f009:**
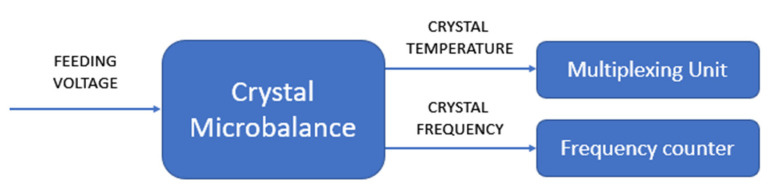
Measurement-chain block diagram.

**Figure 10 sensors-22-07256-f010:**
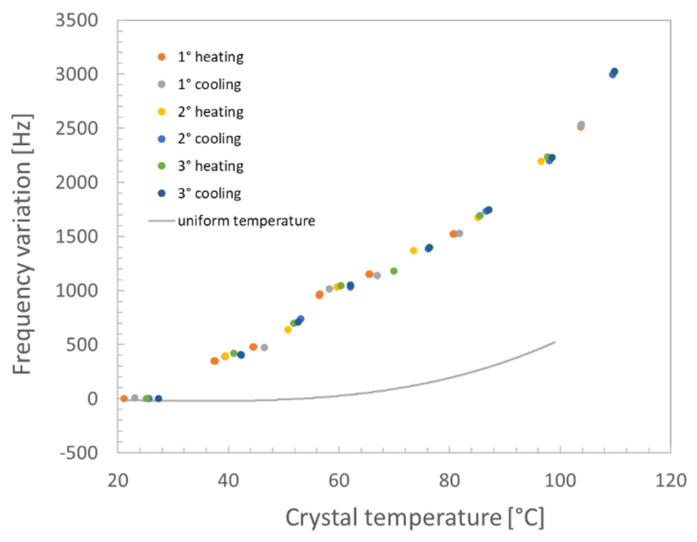
Measured frequency variation with the localized heating of the crystal and the uniform temperature condition.

**Table 1 sensors-22-07256-t001:** Third-order polynomial coefficients from the best calibration data fit.

	$a_{3}$	$a_{2}$	$a_{1}$	$a_{0}$
Units	Hz/°C^3^	Hz/°C^2^	Hz/°C	Hz
Regressor	1.393 × 10^−3^	−0.1012	1.85	−21.2
Standard deviation	2.1 × 10^−5^	3.7 × 10^−3^	0.19	2.9

**Table 2 sensors-22-07256-t002:** Corrected temperatures and measured uncertainties.

	$\bar{T_{bk}}$	$T_{th\;ele}$	$T_{\;ele}$	$T_{\;heater}$	$u_{Fc}$	$u_{T\;bk}$	$u_{\varepsilon }$	$u_{T\;ele}$	$u_{T\;heater}$
Units	[°C]	[°C]	[°C]	[°C]	-	[°C]	-	[°C]	[°C]
Value	23.16	43.16	86.10	90.88	0.0029	0.20	0.0032	0.85	0.42

## Data Availability

Not applicable.

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
