# Peer review of "Characterization of Thermal Gradient Effects on a Quartz Crystal Microbalance"

_sensors, 2022, doi:10.3390/s22197256_

Round 1
Reviewer 1 Report
The paper is devoted to study of temperature gradients on the frequency characteristics of quartz microbalance. The paper and results obtained are interested and could be published after some corrections.
1. The temperature effect on frequency characteristics of QCM is very old problem. It is necessary to show that this problem is still interesting.
I suggest to add in Introduction fresh papers like:
- Huang, Q., Wang, J., Gan, N., et al. An Analysis of the Thermal Behavior and Effects of Circular Quartz Crystal Resonators for Microbalance Applications// IEEE Transactions on Ultrasonics, Ferroelectrics, and Frequency Control, 2022, 69, 2569-2578
- Wu, R., Wang, W., Chen, G., et al. Frequency–Temperature Analysis of Thickness-Shear Vibrations of SC-Cut Quartz Crystal Plates with the First-Order Mindlin Plate Equations//Acta Mechanica Solida Sinica, 2021, 34, 516-526
- Patel, M.S., Sinha, B.K. Pressure and Temperature Sensitivity of a Dual-Mode Quartz Pressure Sensor for High Pressure Applications// IEEE International Ultrasonics Symposium, IUS, 2018, 8579725
2. Provide, please, flow chart and scheme of the experiment in Item 2. Please, point “deposited resistor”. It will be useful for readers.
3. After Fig.1 you use abbreviation LS. What is it? Give explanation what is PT100 in Fig.2a.
4. What is VC in Fig.3?
5. What is BB below Fig.3?
6. What is Th-bb on Fig.4?
Reviewer 2 Report
This research is of great significance to temperature compensation for quartz crystal microbalance or quartz crystal oscillator. The temperature gradient distribution generating by the harmonic vibration of the quartz resonator is considered to be the main reason for the poor temperature repeatability. It is difficult for a temperature sensor at one point to accurately evaluate the resonator temperature plane distribution (or volume distribution). This paper and the previous paper (reference 16)have experimentally confirmed the existence of temperature gradient, and provided a method for measuring temperature gradient, which has reference significance. However, this paper still lacks in theory and data analysis. Some suggestions are as follows:
1. In line 208, in addition to the text description, readers also need a system block diagram to quickly understand the experimental system and the purpose of the experiment.
2. In Figure 7, Is there a quantitative relationship between the frequency repeatability and the observed temperature gradient.
